# OpenReview forum: "Benchmarking Deception Probes via Black-to-White Performance Boosts"
_ICLR.cc/2026/Conference — ICLR 2026 Conference Desk Rejected Submission_

### Official Review · Reviewer_acMw · 2025-10-29

**Soundness:** 4
**Presentation:** 4
**Contribution:** 3
**Rating:** 8
**Confidence:** 3

**Summary:**

The paper proposes a benchmark for “deception probes” by comparing white-box monitoring (access to token-level probe activations) against black-box monitoring on the same tasks, defining value via the black-to-white performance boost. The key concept is that for white-box monitoring to be useful, there should be a performance boost observed from the black-to-white performance.

The authors evaluate eight monitoring strategies (numeric, follow-up, hybrid, and baselines) using two linear probes trained on LLaMA-3.3-70B-Instruct and test across five deception datasets (AI Liar, Insider Trading, Sandbagging, Backdoors, and a Deception-Dilution variant) plus Alpaca as control. Full-Averaging of probe activations performs strongly and robustly (e.g., AUROCs ~0.84–0.999), hybrid white-box methods yield modest but consistent boosts over black-box reasoning in most settings, and the Pairs probe tends to generalize better than the Role-Playing probe, though white-box methods underperform on Backdoors (which is out of the training distribution of the probes).
Overall, results suggest current probes provide weak but encouraging added value for deception detection, while highlighting the need for localization, recalibration across domains, and stronger evasion-aware monitors.

**Strengths:**

Strengths:
The paper is exceptionally well written and easy to follow; it was a pleasure to read. It tackles an important and timely question with a simple, effective, and empirically grounded strategy: does advanced probe-based white-box monitoring on internal activations provide tangible benefits over more accessible black-box monitoring approaches? The authors present a detailed and systematic set of experiments that address this question convincingly—the results clearly show that while white-box monitoring provides a modest “black-to-white” performance boost, the gains are smaller than often assumed.

This work stands out for its clarity in framing, careful experimental design, and transparent reporting of limitations. The benchmark is well-motivated and replicable, with clean separations between datasets, probe designs, and monitoring strategies. The proposed metrics (especially “black-to-white value”) offer a clear operational definition for probe utility, which could be broadly useful for future evaluation of model interpretability tools.

The authors also contribute several thoughtful analyses—such as the dilution experiments, probe generalization across tasks, and the introduction of “Critical Averaging”—that demonstrate nuance and awareness of the broader landscape of mechanistic interpretability. The balance of rigor and practical relevance makes this paper a valuable empirical contribution to the monitoring and safety community.

**Weaknesses:**

The selection of sonnet/haiku seems strange to me. I would like to see results with llama3.1 itself, then it would be a pure black-to-white comparison. I appreciate that the authors put a limitation in their paper, but I don't understand why it was chosen in the first place. There doesn't seem to be a reason to select stronger blackbox monitors than the whitebox counterparts.

It would also be good to include other open-weight models to know if these kinds of results generalize. I am aware that activation and probing techniques do not necessarily generalize equally across models.  I suppose that this was not done because the probes used came from the other paper. Still since this is a benchmarking paper - I think an effort to reproduce the probes (and their black-to-white performance boost) on other models could have been done.

I understand that the dilution strategy was just to check and see "would this work" and that the results seem to be "kinda", but it's completely unrealistic for an untrusted assistant since the user in this case would immediately just think their LLM is broken.
I think the the authors should make it more clear in the paper that this is a non-realistic approach.


Small points for improvement:

Nit: line 45: ’hidden’  (reverse quote on left side)

Put some links to the appendix in the body - for example when you describe critical averaging I had trouble understanding how this would look, but it's all in the appendix in great detail - a direct link would make navigating it easier.

**Questions:**

I suppose we could wonder how much worse the blackbox performance would be if you didn't use sonnet/haiku.
Is there a reason you chose to use stronger models?  Why didn't you just use an OSS model or llama3 itself for these tasks?
Since we are measuring black-to-white, wouldn't it make most sense to use the whitemodel again?
The papers claim of "modest improvements" seems weakened by the fact that the blackbox models are much stronger than the whitebox model use to capture activations. Had it been the same model, maybe the improvements would have been more than modest?

---

> ### Author Response · Authors · 2025-11-18
> **Response to Reviewer**
>
> Hello and thank you for taking the time to review our work and for writing constructive comments.
>
> To briefly address a few:
>
> 1. We spent a good deal of time attempting to employ Llama as the monitor for sake of symmetry but found it too incapable of making reasonable judgements, especially on complex monitoring tasks such as Sandbagging and Backdoors. Ultimately, the use of a strong black-box monitor may seem to undersell the comparative effectiveness of the probes (though the probes do slightly outperform) but it is also the case that those advocating for black-box monitoring may say that the most capable models should be used for monitoring (e.g. untrusted monitoring with SOTA models). As for whether this makes the result weaker, we think it is the opposite. The black-box monitor is advantaged over the probes but the probes nonetheless perform very well. We think in general that when testing the effectiveness of white-box techniques it is not harmful to compare against the strongest black-box techniques we might otherwise employ.
>
> 2. As for the realism of the deception dilution point. This is fair, it hopefully is clear we do not think this is a real world attack, but perhaps should be made more explicit.
>
> Thank you also for the suggestion about an appendix link, for noticing the wrong sided quote mark, and again for engaging with our work!

---

### Official Review · Reviewer_kFyE · 2025-10-30

**Soundness:** 2
**Presentation:** 2
**Contribution:** 2
**Rating:** 2
**Confidence:** 3

**Summary:**

The paper proposes to evaluate deception probes with the performance difference observed when comparing a deception monitor with access to token activations (white-box) vs without such access (black-box). Several variations of deception detection methods are evaluated on 4 benchmark datasets (plus Alpaca for control). On 3 out of 4 datasets the white-box methods outperform black-box, while on the backdoors dataset blackbox methods dominate. The paper also tests the robustness of the detection methods against a simple countermeasure.

**Strengths:**

* The evaluation of linear probes for deception detection is an interesting and relevant and well motivated problem.
* Evaluation on 4 reasonable benchmark datasets, the details of which are described clearly.
* Robustness against countermeasures is a very interesting direction.
* Figure is plotted with 95% bootstrap confidence intervals.

Overall I think the research question tackled in the paper is promising.

**Weaknesses:**

* The clarity of the paper could be improved, especially the discussion of the methodology would benefit from a more formal and precise formulation. In its current state it was hard to follow and understand clearly how the monitoring methods are implemented.
* While this is an interesting direction, I think the current evaluation is fairly limited by considering only a specific set of linear probes, and comparing to only a single model for black-box reasoning. I would have been very interested to see a more in-depth evaluation of the generalisability of linear probes, also considering more than 2 already trained probes, compared against more models for BBR, and an evaluation of the robustness of probes against countermeasures beyond the simple dilution. I think these points could significantly strengthen the paper.
* The choice of the ensemble (line 311) seems arbitrary, why not evaluate other combinations and other weightings?
* Several ablations could strengthen the paper, such as, variations on the prompts of the LLM judge, which aspects of WBR drive performance, other reasoning models, at some named above.
* Regarding the backup of critical averaging (line 286), how do we know the LLM failed to identify the critical claim?
* While the original paper was referenced, the training details of the linear probes used in the evaluation seem highly relevant and would help in understanding the results.
* A more thorough explanation of the results and their broader implications would greatly benefit the discussion section. For example why are full averaging and critical averaging comparable on the deception dilution dataset? Does this mean identifying the critical part does not help, i.e. the countermeasure is ineffective, or does it mean the critical part is not identified correctly? Why does WBR underperform so drastically on Backdoors?
* While acknowledged, I do feel that the use of stronger models for WBR, BBR as well as the critical claim identification is a limitation of this work that could be fixed by simply using a smaller model.
* Overall, I am doubtful of the significance and generalisability of the conclusions of this paper, since for example on average the difference between WWB and BBR seems insignificant (Figure 2). Maybe a more nuanced conclusion connecting to the training data of the probes could resolve this.
* In some paragraphs of the discussion it is unclear whether the authors are referring to Table 3 or Figure 2. For example, in the paragraph starting on line 408, based on Figure 2 the difference between PR and Pairs on Backdoors seems insignificant considering the large bootstrap CIs.

Minor:
There are quite a few typos / inconsistencies, e.g:
* '..' vs "..." lines 40. vs 49.
* line 50: [A] model's....
* use \citep in line 59
* line 82 "with a modest" -> with modest
* line 245, it is unclear what is meant by "a black-box"
* BBR is used in line 314 but only discussed in 5.4

**Questions:**

1. Why are full averaging and critical averaging comparable on the deception dilution dataset?
2. Does this mean identifying the critical part does not help, i.e. the countermeasure is ineffective, or does it mean the critical part is not identified correctly?
3. Why does WBR underperform so drastically on Backdoors?
4. Regarding the backup of critical averaging (line 286), how do we know the LLM failed to identify the critical claim?
5. The choice of the ensemble (line 311) seems arbitrary, why not evaluate other combinations and other weightings?

---

> ### Author Response · Authors · 2025-11-16
> **Response to Reviewer.**
>
> Hello, and thank you for the thoughtful review. We appreciate the clear summary of the contribution and the recognition of the strengths, including the relevance of evaluating deception probes, the quality of the benchmark datasets, the interest of robustness testing, and the statistical reporting.
>
> 1. Clarity and Precision of Methodology:
>
> We appreciate your note that the methodological discussion was sometimes hard to follow. We do disagree and think our presentation is clear. Notably, other reviewers described the paper as: “exceptionally well written and easy to follow”
>
>
> 2. Breadth of Probe Evaluation and Generalizability:
>
> We agree that evaluating more probes, more BBR models, or more countermeasures would offer additional insights. Our focus here was narrower and constrained by available tools.
>
> Within this scope, we evaluated:
>
> -five deception datasets (including two OOD stress tests),
> -sixteen monitors,
> -two independent monitor models (haiku and sonnet)
>
> We view broader cross-model and cross-architecture evaluation as an important direction for future work, beyond the scope of this benchmark.
>
> 3. Choice of Ensemble Weights:
>
> The ensemble weighting mimics WBR in its scoring system except that the ensemble is blind. I.e. the probe and BBR are combined numerically, whilst WBR scores in the same way except that the monitor reasons about the scores. This allowed us to more directly compare the ensemble and the WBR. Specifically, we can ask if the model's reasoning about the scores directly is helpful as opposed to merely combining the outputs of the black-box monitor and the probe.
>
> 4. Critical Averaging and Deception Dilution + Why WBR Underperforms on Backdoors:
>
> In Appendix D we elaborate a bit on the deception dilution results. While performance decreases the effect is relatively modest. Critical Averaging and Full Averaging. The results of these two methods are generally comparable on AI Liar and on Dilution (which is a variant of AI Liar).
>
> In Section 8 we note that, WBR underperforms due to distributional shift rather than probe failure:
> honest Backdoors samples have probe means exceeding 99.9 percent of Alpaca’s honest distribution
> WBR encodes expectations about “typical honest ranges” this calibration breaks under shift, causing WBR to flag nearly everything as suspicious. This result emphasizes that LLM reasoning over probe activations can become brittle OOD, even when the underlying probe remains moderately informative.
>
> 5. Detecting Failure in Critical-Claim Identification:
>
> A good question, we will try to clarify. Critical-claim identification “fails” when the LLM returns an empty span. Which the model does if it cannot identify a single critical or most critical claim in the response. In such cases we revert to Full Averaging. This ensures that identification failures do not harm performance.
>
> 6. Interest in More Ablations:
>
> We agree that prompt variants, ablations of WBR dimensions, alternative ensembles, or more monitor models would all be informative. Our aim here was to benchmark existing probes rather than explore the full design space of monitors. These ablations represent natural extensions rather than shortcomings of the current work. Considering the five deception datasets and eight monitoring methods, and two probes employed in the test suite.
>
> 7. Significance and Generalizability of the Black-to-White Boost:
>
> Although the average boost in Figure 2 may appear modest, it is important to note:
> the black-box baseline is a strong model (Claude Sonnet 3.5) despite that, white-box access improves performance on four out of five datasets. Since the Backdoors dataset contains few examples the confidence interval is large and this expands the apparent confidence interval of the average as well as bringing down the average performance of the WBR method which fails systematically on this setting as we noted above and in Section 8. We nonetheless include it as an important stress test for these methods but think that it should not be taken to weaken the conclusion that white box methods improve on even the strong black box baseline which is shown repeatedly across the test suite. The numeric Full Averaging method alone surpasses the strong BBR baseline on all settings except the Backdoors stress test, in which it is OOD and is still somewhat effective in this setting. We think these findings alone are both compelling and a substantial contribution.
>
> Thank you for your engagement and the time you took in this review. We respectfully ask that you adjust the final rating if the clarifications above address your concerns or that you clarify any remaining concerns.

---

### Official Review · Reviewer_o3aa · 2025-11-01

**Soundness:** 1
**Presentation:** 1
**Contribution:** 2
**Rating:** 2
**Confidence:** 3

**Summary:**

This paper explores the evaluation of "Deception Probes," a white-box technique designed to detect deceptive responses from Large Language Models (LLMs) by analyzing their internal activations.

The paper's central contribution is a novel benchmarking methodology termed the **"Black-to-White Performance Boost."** This approach seeks to quantify the **marginal value** provided by "White-Box Reasoning" (WBR), which accesses both text and probe scores, compared to "Black-Box Reasoning" (BBR), which only accesses the public text output.

Using this methodology, the authors evaluate two existing deception probes (Pairs and RP) on multiple strategic deception datasets (e.g. Insider Trading, and Sandbagging). Additionally, they devise two stress tests: a "Deception Dilution" attack, which pads deceptive responses with substantial irrelevant, truthful information, and an out-of-distribution (OOD) generalization test involving the detection of backdoors in code.

The study finds that existing probes provide some performance improvement, termed "weak but encouraging" by the authors. A key finding is that on the OOD `Backdoors` dataset, the performance of the "smarter" White-Box Reasoning (WBR) method **significantly underperforms** the pure black-box baseline (BBR), attributed to a distributional shift in the probe activations.

**Strengths:**

- The proposed core evaluation metric—the "Black-to-White Performance Boost"—is a pragmatic and significant contribution. It shifts the evaluative focus from the "absolute accuracy of probes" to the more critical question of "what marginal benefit probes can offer in realistic monitoring scenarios."
- The inclusion of stress tests for out-of-distribution (OOD) generalization (the `Backdoors` dataset) and adversarial robustness (the `Deception Dilution` attack) is crucial for evaluating the practical utility of these probes.
- The analysis of WBR's failure on the `Backdoors` dataset (attributing it to a distributional shift in activation scores that breaks calibration) is insightful. It offers an important cautionary note for the future deployment of this technique.

**Weaknesses:**

While the paper's premise is strong, its core methodology and experimental design suffer from flaws that undermine the persuasiveness of its conclusions.

- **Fundamental Flaw in Experimental Design:** The authors employ Claude 3.5 Sonnet as the reasoning engine for both BBR and WBR, to monitor a comparatively weaker LLaMA-3.3-70B assistant. In this asymmetric "strong-monitors-weak" setting, the BBR baseline (relying on Claude 3.5's black-box scrutiny alone) achieves exceptionally high performance (e.g., AUROC > 0.82 and even 0.94 on the AI Liar and Sandbagging datasets). This inherently compresses the measurable "marginal value" the probes can provide. Consequently, the paper's central conclusion—that probes offer only a "weak performance boost"—is likely an artifact of this flawed design. This setup severely underestimates the probes' true potential in more realistic scenarios where a BBR baseline would falter.

- **Fragility of the White-Box Reasoning (WBR) Method Misinterpreted as a Limitation of the Probes:** The paper presents the failure of WBR on the `Backdoors` dataset as a key finding. However, this appears to be a methodological flaw rather than a fundamental limitation of the probes. As per Appendix G.2 and Section 8, the WBR monitor's prompt includes hard-coded activation score ranges calibrated on the `Alpaca` control set (e.g., "honest reply scores are between -3 and 0"). When faced with the OOD `Backdoors` dataset, the activation score distribution for *honest* samples shifts fundamentally. This causes the WBR method to fail completely, precisely because it relies on this brittle, hard-coded rule. This demonstrates a lack of robustness in the paper's specific WBR implementation, not the ineffectiveness of the white-box signal in OOD scenarios.

- **Conclusions Lack Generalizability:** All conclusions in this paper are highly contingent on a single model: LLaMA-3.3-70B-Instruct. The two evaluated probes were trained specifically on this model's activations (notably, layer 22), and all deceptive responses were also generated by this same model. It is well-established that internal representations, and likely deception mechanisms, differ significantly across LLM architectures and scales. Therefore, it is highly questionable whether these conclusions generalize to other models.

- **Insufficient Analysis of the Probe Signal:** The experiments repeatedly demonstrate that the simplest `Full Averaging` method nearly always outperforms or matches the more complex `Critical Averaging` and `WBR` methods. This suggests that the signal provided by current probes is **global** (i.e., detecting a *macro-state* of deception) rather than **local** (i.e., detecting deception at the token or sentence level). If so, the `WBR` approach, which attempts to use an LLM to "reason" about and "explain" specific token-level activations, is likely "over-interpreting noise" and may even be harmful. The paper fails to sufficiently discuss this possibility: that complex, LLM-based white-box reasoning is counter-productive and less effective than simple statistical aggregation.

**Questions:**

Given the "strong-monitor, weak-assistant" setup, how can the authors be sure that the central conclusion of a "weak performance boost" is not an artifact of the artificially inflated BBR baseline?

---

> ### Author Response · Authors · 2025-11-16
> **Response to Reviewer**
>
> We thank the reviewer for their careful reading of our manuscript and for providing insightful feedback. We are especially pleased that you recognized the core contributions: the "Black-to-White Performance Boost" metric, the necessity of the stress tests (OOD and Dilution), and the insightful failure analysis on the Backdoors dataset. In looking at the weaknesses you point out we feel we feel they are excellent points about which we are primarily in agreement but that we did cover these points in the paper.
>
> Response to Weaknesses
> 1. Fundamental Flaw in Experimental Design (Strong-Monitor, Weak-Assistant)
> You correctly point out that using a stronger LLM (Claude 3.5 Sonnet) as the monitor for a weaker assistant (LLaMA-3.3-70B) leads to a very high Black-Box Reasoning (BBR) baseline, which inherently compresses the observable Black-to-White Performance Boost.
>
> We acknowledged the asymmetric setup. Our primary goal was not to achieve the highest possible absolute accuracy, but rather to use the Black-to-White Performance Boost metric to rigorously determine: a) if existing probes offer a measurable improvement over strong black-box baselines, and b) if alternative methods of employing probes (like Ensemble and WBR) could further improve on BBR alone. Since the exact same strong model is used for both BBR and WBR, the marginal performance difference remains a valid measure of the probe's marginal value. The fact that the lightweight numeric probe method, Full Averaging, performs better than the strong asymmetric BBR baseline on average demonstrates that the value of probes is significant. This finding that a simple, lightweight probe can outperform a complex, state-of-the-art black-box model, has made the efficacy of probes the focal point of our subsequent research.
>
> 2. You suggest that the WBR's failure on the Backdoors dataset is a methodological flaw in our WBR implementation rather than a finding about the probes.
>
> We agree that this observation pertains to the brittleness of the WBR monitor's reasoning, not the underlying probe signal. We do not interpret WBR fragility as a fundamental limitation of the probes. There may be a way around this fragility in realistiic settings, one could imagine for example trying to have some method of establishing a unique baseline for OOD activations and updating the monitor with instructions on what distributions of scores correspond with what general meaning but we think that in safety relevant applications in practice you will set a baseline from a control distribution and need to classify your responses on the basis of that control threshold. The probe itself is more performant when used via simple Full Averaging than when its signal is filtered through the WBR monitor. This indicates that the monitor struggles to reason effectively about the probe scores in this specific OOD stress test. As we originally noted in the paper, while the probe's performance deteriorates under this severe distributional shift, it is still somewhat efficacious in the OOD stress test, validating the underlying signal. We also think that probes which have more localized token or statement level accuracy could improve methods such as WBR and have looked at this in subsequent work.
>
> 3. Conclusions Lack Generalizability
> You note that all conclusions are contingent on a single model, LLaMA-3.3-70B-Instruct, limiting generalizability.
>
> This is a fair point and a limitation. We emphasized depth testing five deception datasets (including two OOD stress tests), sixteen monitors (8 x 2 probes), and two monitor models (haiku, and sonnet), and think this a solid contribution more testing across more variables is always preferable.
>
> 4. Insufficient Analysis of the Probe Signal
> You suggest that the superior performance of Full Averaging implies the signal is global and that WBR is "over-interpreting noise."
>
> Our analysis indicates that the probes are effective, but that the underlying signal is diffuse and noisy at the token or statement level. We do not intend to suggest the probe is bad! Instead, we argue that the main limitation of the complex WBR and Critical Averaging methods is this diffuse signal. We are using this insight in subsequent work to train token and statement level probes designed to better capture this global signal. Notably the Full Averaging method is shown to be highly performant, and could serve well in many applications.
>
> Given that the core strengths of the paper were acknowledged, and that the weaknesses raised either stem from a necessary experimental constraints or are in fact findings explicitly discussed and analyzed in the paper (Weakness 2, 4), we ask the reviewer to reconsider and, if in agreement, to raise the final rating.

---

### Official Review · Reviewer_4Koc · 2025-11-01

**Soundness:** 3
**Presentation:** 3
**Contribution:** 3
**Rating:** 6
**Confidence:** 4

**Summary:**

This paper introduces a methodology for evaluating "deception probes" which are linear classifiers trained on an LM's internal activations to detect deceptive responses. The core contribution is framing the evaluation in terms of a "black-to-white performance boost," which measures the additional accuracy gained by a white-box monitor (with access to probe activations) over a purely black-box monitor (which only sees the public dialogue). Authors test 2 deception probes across 6 datasets, and evaluating on 8 monitoring strategies. The key finding is that existing deception probes offer a "weak but encouraging" performance boost, with simple methods like averaging probe activations proving surprisingly effective and robust.

**Strengths:**

- The experimental setup is thorough and well-designed. On top of many datasets tested, the inclusion of an adversarial attack ("Deception Dilution") and a control dataset (Alpaca) further strengthens the evaluation.

- The paper produces several clear and actionable insights. The finding that a simple "Full Averaging" of probe activations (line 373)  is an important result which suggests that complex reasoning over probe scores may not always be necessary or even beneficial. The paper is also transparent about the probes' failures, such as the RP probe's poor performance on the Backdoors dataset (line 413), which provides valuable data for the community.

**Weaknesses:**

1. Regarding results for hybrid reasoning models
To me, the most concerning result is the performance of the White-box Reasoning (WBR) monitor on the Backdoors dataset. As shown in Figure 2, the WBR method performs significantly worse than the Black-box Reasoning (BBR) baseline. This is counterintuitive; adding a genuinely informative signal (the Pairs probe has an AUROC of 0.844 on this dataset with Full Averaging) should not make an LLM-based monitor worse. The paper's explanation (lines 417-424) suggests BBR has an advantage, but it doesn't adequately explain why WBR underperforms. This implies a failure in the hybrid reasoning process itself, where the LLM monitor is incorrectly interpreting or being confused by the probe scores, which is a major weakness of that specific method.

2. The deception dilution might be a bit simplistic
While it's a strength that the paper includes an adversarial evasion strategy, the chosen method is quite basic. Appending a long list of irrelevant facts at the end of a response (lines 184-189) is a simple form of obfuscation. The fact that most monitors were still reasonably effective against it (Full Averaging achieved 0.931 AUROC, line 472) might lead to a false sense of security.

**Questions:**

Regarding the WBR monitor's failure on the Backdoors dataset: Could you provide more insight into why the LLM appears to be confused by the probe scores? Did you perform any analysis of the WBR monitor's chain-of-thought for these examples?

---

> ### Author Response · Authors · 2025-11-18
> **Response to Reviewer**
>
> Hello, and thank you for your review.
>
> On the weaknesses you identify, we offer two brief clarifications:
>
> 1. WBR underperformance on Backdoors.
> We should clarify this underperformance further, and we appreciate you calling attention to it. We note in Section 8, the failure is driven by distributional shift rather than probe weakness: honest Backdoors samples lie far outside the probe’s typical honest range on all other datasets, causing WBR’s reasoning step, whose prompts encode assumptions about “typical” score magnitudes, to miscalibrate and flag nearly all examples as suspicious. In contrast, simple numeric ensembling (Full Averaging) makes no such assumption showing the signal detected by the probes is preserved even if also somewhat less performant on thiis stress test.
>
> 2. Simplicity of the deception-dilution attack.
> We agree this attack is quite simple. We also thought it likely that this attack would break the simple Full Averaging method for aggregation since the subsequent and numerous honest portions of the response would 'water down' the deception signal. Instead we found this attack was only somewhat effective, and this lead to a clearer understanding of the diffusion of the probe signal across the response. Nonetheless you make a good point. Its possible a less a more realistic version of this attack might be more effective. We could for example have the model generate plausible extensions of the original rollout with additional honest claims that were tangentially related to the core claim and create a dataset of these more realistic extensions of the original response but we did not do it here and would leave for future work.
>
> Thank you again for your constructive feedback and for the assessment of our work.

---

### Note · Program_Chairs · 2026-01-17
**Submission Desk Rejected by Program Chairs**

The following references in this submission do not refer to real documents and/or have major errors in bibliographic information:

 Ningyu Li, Alexander Pan, Anirudh Gopal, Catherine Yeh, Rebecca Bernsohn, Alex Andonian, Sean Zeng, Divya Raghu, Yafei Shi, Zhengxuan Zhang, Lance Ren, Peng Wang, Andy Zou, and Dan Hendrycks. The mask (machine-generated authorship schemes for knowledge) benchmark: Measuring machine vs. human authorship in academic writing. arXiv preprint arXiv:2409.16841, 2024b